# Changes in Rumen Microbiology and Metabolism of Tibetan Sheep with Different Lys/Met Ratios in Low-Protein Diets

**DOI:** 10.3390/ani14111533

**Published:** 2024-05-22

**Authors:** Fengshuo Zhang, Yu Zhang, Tingli He, Qiurong Ji, Shengzhen Hou, Linsheng Gui

**Affiliations:** College of Agriculture and Animal Husbandry, Qinghai University, Xining 810000, China; zhangfengshuo1997@163.com (F.Z.); iszhangyu06@163.com (Y.Z.); j1960742393@163.com (Q.J.); qhdxhsz@163.com (S.H.)

**Keywords:** low-protein diets, amino acids, *Ovis aries*, antioxidants, digestive enzymes, rumen microbes, off-target metabolomics

## Abstract

**Simple Summary:**

The first and second limiting amino acids in ruminants 66 are methionine and lysine, respectively, and an appropriate lysine-to-methionine ratio is necessary for amino acid balance in a low-protein diet. A Lys/Met ratio of 1:1 increased the antioxidant capacity and the activities of digestive enzymes and reduced the production of ammonia nitrogen in Tibetan sheep. In addition, 16S rDNA sequencing revealed that a Lys/Met ratio of 1:1 significantly increased the Ace and Chao1 indices, and non-target metabolomics analysis revealed that cis-jasmone and Val-Asp-Arg could be involved in mediating the antioxidant capacity and increasing the activity of digestive enzyme activities in Tibetan sheep rumen. Phosphoric acid, one of the metabolic products, increased cellulase activity by regulating the abundance of Succiniclasticum through the oxidative phosphorylation pathway.

**Abstract:**

In ruminants, supplementing appropriate amounts of amino acids improves growth, feed utilization efficiency, and productivity. This study aimed to assess the effects of different Lys/Met ratios on the ruminal microbial community and the metabolic profiling in Tibetan sheep using 16S rDNA sequencing and non-target metabolomics. Ninety-two-month-old Tibetan rams (initial weight = 15.37 ± 0.92 kg) were divided into three groups and fed lysine/methionine (Lys/Met) of 1:1 (LP-L), 2:1 (LP-M), and 3:1 (LP-H) in low-protein diet, respectively. Results: The T-AOC, GSH-Px, and SOD were significantly higher in the LP-L group than in LP-H and LP-M groups (*p* < 0.05). Cellulase activity was significantly higher in the LP-L group than in the LP-H group (*p* < 0.05). In the fermentation parameters, acetic acid concentration was significantly higher in the LP-L group than in the LP-H group (*p* < 0.05). Microbial sequencing analysis showed that Ace and Chao1 indicators were significantly higher in LP-L than in LP-H and LP-M (*p* < 0.05). At the genus level, the abundance of *Rikenellaceae RC9 gut group* flora and *Succiniclasticum* were significantly higher in LP-L than in LP-M group (*p* < 0.05). Non-target metabolomics analyses revealed that the levels of phosphoric acid, pyrocatechol, hydrocinnamic acid, banzamide, l-gulono-1,4-lactone, cis-jasmone, Val-Asp-Arg, and tropinone content were higher in LP-L. However, l-citrulline and purine levels were lower in the LP-L group than in the LP-M and LP-H groups. Banzamide, cis-jasmone, and Val-Asp-Arg contents were positively correlated with the phenotypic contents, including T-AOC, SOD, and cellulase. Phosphoric acid content was positively correlated with cellulase and lipase activities. In conclusion, the Met/Lys ratio of 1:1 in low-protein diets showed superior antioxidant status and cellulase activity in the rumen by modulating the microbiota and metabolism of Tibetan sheep.

## 1. Introduction

Reducing the amount of protein in the diet could decrease nitrogen emissions and feed costs. The connection between increasing the amount of amino acids in feeds to meet animal growth requirements and balanced amino acids in the body is not clear [1]. The first and second limiting amino acids in ruminants are methionine and lysine, respectively, and an appropriate methionine-to-lysine ratio is necessary for amino acid balance in a low-protein diet [2]. Thus, all proteinogenic amino acids are physiologically and nutritionally required for animal protein synthesis, growth, development, and good health.

Appropriate dietary amino acid supplementation enhances animal growth and ensures better feed utilization efficiency and high productivity [3,4]. According to Kamiya et al. [5], dietary lysine and methionine supplementation reduce nitrogen excretion in fattening Holstein steers without affecting productivity [6]. Dietary supplementation with 15 g/d of methionine and sodium nitrate altered the abundance and composition of rumen microbes in buffaloes [7]. In Hu Sheep, the dietary supplementation with methionine increased the apparent digestibility via altering ruminal fermentation characteristics and consequently promoted the dry matter intake and average daily gain [8]. Moreover, supplementing the methionine (6 g/animal/day) in ewe’s diet improved the oxidative status in the serum, as well as the chemical composition in milk [9]. However, there is limited information on the impact of dietary supplementation with different Lys/Met ratios in low-protein diets on ewes’ oxidative status, bacterial community, and metabolic profiling in the rumen.

Excessive free radicals harm cells and macromolecules like proteins and nucleic acids, altering the stability of an organism’s internal environment. Oxidative stress can impact animal health, causing a number of disorders [10]. The antioxidation levels are a crucial defense against external stimuli, regulate response to infection, and are a critical indicator of an animal’s overall health. T-AOC, CAT, GSH Px, SOD, and MDA levels are the main antioxidant level markers in animals. Variations in these markers’ content indirectly indicate the body’s antioxidant capacity and antioxidant activity under oxidative stress [11].

With a high diversity of microorganisms, the rumen of Tibetan sheep is termed an anaerobic fermentation chamber [12]. Feed digestibility and rumen environmental homeostasis are highly dependent on the abundance and diversity of rumen microbes [13]. As a valuable tool for identifying the structure and composition of the microbial community, 16S rDNA is thought to be the most appropriate indicator of bacterial phylogeny [14,15].

Non-target metabolomics detects numerous metabolites in an unbiased, large-scale, and systematic manner. This technology was used to determine distinct metabolites and reveal the correlation between nutrition intake and the function of the gastrointestinal tract [16]. Amino acids are mostly produced in the rumen through the breakdown of food protein and bacterial protein by rumen microbiota. Numerous metabolic pathways are involved in the production of amino acids, the essential building blocks of proteins and peptides. Variations in metabolites produced may be influenced by the internal environment and gene expression levels of rumen bacteria [17].

As a key source of income for farmers on the Qinghai–Tibet Plateau, Tibetan sheep (*Ovis aries*) could survive at 3000 m above sea level. The sheep are well adapted to extreme cold, hypoxia, and starvation [18]. In the present study, the effect of different Lys/Met ratios in low-protein diets on the composition of the rumen microbes and fermentation characteristics in the rumen was investigated using gas chromatogenic-mass spectrometry (GC-MS) and 16S rDNA sequencing. Non-target metabolome analysis was used to determine the metabolites produced by rumen bacteria.

## 2. Materials and Methods

### 2.1. Experimental Design and Sample Collection

The experiment was carried out for 100 days, including 10 days of feed acclimatization and 90 days of actual experiment. The feeding test was conducted on a commercial farm (Haibei Tibetan Autonomous Prefecture, Haibei, China). Ninety healthy, two-month-old Tibetan plateau rams (body weight = 15.37 ± 0.92 kg) were randomly selected for the experiment. The rams were randomly split into three groups, each containing thirty rams, and housed in separate pens (30 Tibetan sheep per pen). A low-protein diet (approximately 10%) contained 30% roughage (oat silage and oat hay to dry matter ratio of 1:1) and 70% concentrate. The three treatment groups were fed the diets with Lys/Met ratios of 1:1 (LP-L), 2:1 (LP-M), and 3:1 (LP-H), respectively. Additionally, soybean meal, rapeseed meal, cottonseed meal, maize germ meal, and palm meal were used as the protein source to provide 12% crude protein in the diet. Table 1 shows the composition and nutritional content of the diet. At the end of the feeding test, six sheep were randomly selected from each group, held off feed for 24 h, and denied water for two hours before being slaughtered. The rumen fluid was collected and stored at −80 °C until analysis. The slaughtering and sampling were performed in accordance with standard procedures.

### 2.2. Chemical Analysis

Crude protein (No. 984.13) and ether extract (EE) (No. 920.39) were measured as proposed by the Association of Official Analytical Chemists [19]. Neutral detergent fiber was analyzed according to the methods of Licitra et al. (1996) [20]. Acid detergent fiber (ADF) was determined using the methods described by Van Soest et al. (1991) [21]. Ca and P were analyzed using the dry ash method (No. 927.02) and the photometric method (No. 965.17), respectively [19].

### 2.3. Determination of Antioxidant Indicators and Digestive Enzyme Activity in Rumen Fluid

The antioxidant capacity and the activities of digestive enzymes in the rumen fluids were assessed using a one-step sandwich enzyme-linked immunosorbent assay (ELISA). The test kits were purchased from Jiangsu Enzyme Biotech Co., Ltd. (Nanjing, China) and used according to the manufacturer’s guidelines. Briefly, the rumen fluids were transferred to polypropylene microtubes and centrifuged at 3000× *g* for 20 min at 4 °C. Then, 10 µL of the supernatants were removed and mixed with 40 µL of diluent. The mixture was vortexed for 10 s and incubated for 30 min at 37 °C. The liquids were removed and then added 50 µL of enzyme-labeled reagent in 96-well plates. The 50 µL of color-developing agent color-developing agent were vortexed for 10 min at 37 °C. After the termination reaction, the absorbance value was measured under 450 nm wavelength by an enzyme-labeling analyzer (A51119600C, Thermo Fisher, Wilmington, DE, USA).

### 2.4. Determination of Rumen pH and Volatile Fatty Acids Concentration

After slaughtering the Tibetan sheep, the rumen fluids were collected, filtered through 4 layers of coarse gauze with a mesh size of 250 mm, and stored in a 5 mL test tube. The pH was measured using a portable pH meter (PHB-1, Shanghai, China) according to the manufacturer’s instructions. Approximately 1 mL of rumen fluid containing 0.25 mL metaphosphoric acid (25 g/100 mL) was centrifuged (at 12,000× *g* for 15 min at 4 °C), and the volatile fatty acids (VFA) were analyzed using a gas chromatography equipped with a 30 m × 0.33 mm × 0.25 µm capillary column (GC-2014, Kyoto, Japan). Samples were run at a split ratio of 20:1 under a column temperature of 60 °C to 190 °C, increased at a rate of 20 °C/min, with a 3 min hold at each interval.

### 2.5. Rumen 16S rDNA Gene Sequencing Analysis

#### 2.5.1. DNA Extraction and Sequencing

The genomic DNA of the rumen microbiota was extracted, and its concentration and purity were determined. PCR amplification was performed, and the quality of the PCR products was assessed using 2% agarose gel electrophoresis. Aliquots of the samples were pooled, purified, and sequenced on an Illumina NovaSeq6000 sequencer (Illumina, San Diego, CA, USA) using PE250 read lengths.

#### 2.5.2. Sequencing Data Analysis

Clean sequence reads were clustered using Uparse (http://drive5.com/uparse, accessed on 12 November 2023). Sequences with 97% similarity were grouped into OTUs, and representative sequences from the OTUs were compared to the matching reference data using the RDP classifier algorithm for species annotation. The 16S/18S UNITE database (https://unite.ut.ee, accessed on 25 November 2023) was used for ITS annotation, while the Silva 132 database (https://www.arb-silva.de, accessed on 17 December 2023) was used for sequence annotation. Species richness and diversity were evaluated by standardizing all sample sequences to an equal level (sample with the fewest sequences).

### 2.6. Analysis of Non-Target Metabolomics in the Rumen

#### 2.6.1. Extraction of Metabolites

Briefly, 400 μL of methanol/acetonitrile/H_2_O (2:2:1, *v*/*v*/*v*) was added to 100 mg of the well-vortexed rumen content sample. The mixture was vortexed and centrifuged at 14,000× *g* for 20 min at 4 °C. The supernatant was collected and dried in a vacuum centrifuge at 4 °C before 20 min incubation on ice. The samples were dissolved in 100 μL of acetonitrile/water (1:1, *v*/*v*) solvent and aliquoted into LC vials for Liquid chromatography-mass spectrometry (LC-MS) analysis.

#### 2.6.2. Liquid Chromatography-Mass Spectrometry Analysis

The extracts were analyzed by quadrupole time-of-flight mass spectrometry (SCIEX, Framingham, MA, USA). Liquid chromatography separation was performed in an ACQUIY UPLC BEH Amide column (2.1 mm × 100 mm, 1.7 µm particle size, Waters, Milford, MA, USA) using solvent A (25 mM ammonium acetate and 25 mM aqueous ammonium hydroxide) and B (acetonitrile).

### 2.7. Statistical Analysis and Correlation Analysis

Data were analyzed using SPSS version 26.0. Continuous normally distributed data were expressed as mean ± standard error of the mean (SEM). The KS normality test was performed to estimate data normality. Differences between groups were analyzed using the one-way ANOVA test, while differences between multiple groups were analyzed using Duncan’s test. Statistical significance was set at *p* < 0.05. The association between the rumen bacterial diversity, the abundance of metabolites, the activities of antioxidant enzymes, and fermentation parameters in Tibetan sheep was investigated using Spearman’s correlation coefficient (*p* < 0.05, R > 0.60).

## 3. Results

### 3.1. The Antioxidant Indicators

The contents of antioxidant markers T-AOC, GSH-Px, and SOD were significantly higher in the LP-L group than in the LP-H and LP-M groups (*p* < 0.05). However, no significant difference was observed for these markers between the LP-H and LP-M groups (*p* > 0.05). Additionally, no significance was observed in CAT and MDA between the three groups (*p* > 0.05) (Table 2).

### 3.2. The Digestive Enzyme Activity

A significant difference was observed in the cellulase activity between the LP-L group and the LP-H group (*p* < 0.05). Although the difference was not significant, the α-amylase, trypsin, and lipase activities were higher in the LP-L group than in the LP-H and LP-M groups (*p* > 0.05) (Table 3).

### 3.3. The Rumen Fermentation Parameters

Results for the rumen fermentation indicators are shown in Table 4. The acetate level was significantly higher in the LP-L and LP-M groups than in the LP-H group (*p* < 0.05). Compared with the LP-H and LP-M groups, the ammonia nitrogen level in the LP-L group was significantly lower (*p* < 0.05). No significant difference was observed in propionate, butyrate, isobutyrate, valerate, isovalerate, and pH between the three groups (*p* > 0.05).

### 3.4. Sequence Analysis for 16S rDNA Gene of Rumen Microbiota

#### 3.4.1. Rumen Microbial Abundance and Diversity

As shown in (Figure 1A), 1465 OTUs were identified across the three groups, of which 675, 358, and 432 specific OTUs were identified in the LP-L, LP-M, and LP-H groups, respectively. The species information contained in the samples (Figure 1B) was presented as points in the multidimensional space, and the distance between the points reflected the degree of difference between the different samples. β-diversity analysis (Figure 1C) showed that the Anosim R-value was 0.327, indicating no differences between the three groups. The α diversity index analysis (Figure 1D–F) revealed that the Ace and Chao1 indicators were significantly higher in the LP-L group than in the LP-H and LP-M groups (*p* < 0.05). No significant differences were observed in the Shannon indicator between the three groups (*p* > 0.05).

#### 3.4.2. Relative Abundance and Functional Items at the Phylum and Genus Level of Rumen Microbiota

The top 3 predominantly bacteria phyla were *Firmicutes*, *Bacteroidetes*, and *Proteobacteria* (Figure 2A). The abundance of the *Firmicutes* (Figure 2B) was significantly higher in the LP-L and LP-M groups than in the LP-H group (*p* < 0.05). The abundance of *Bacteroidetes* (Figure 2C) was significantly higher (*p* < 0.05) in LP-L and LP-H groups than in LP-M group. The abundance of *Proteobacteria* (Figure 2D) was significantly higher (*p* < 0.05) in the LP-M group than in the LP-L and LP-H groups. The abundance of *Acidobacteria* (Figure 2E) was significantly higher in the LP-L group than in the LP-M and LP-H groups (*p* < 0.05). At the genus level, *uncultured rumen bacteria Prevotella 1* and *Rikenellaceae RC9* were the most abundant across the three groups (Figure 2F). The abundance of *uncultured rumen bacteria* (Figure 2G) was significantly higher in the LP-H group than in the LP-M group (*p* < 0.05). No significant significance was observed in the abundance of *Prevotella 1* (Figure 2H) (*p* > 0.05) between all three groups. The abundance of the *Rikenellaceae RC9 gut group* (Figure 2I) was significantly higher in the LP-L and LP-H groups than in the LP-M group (*p* < 0.05). The *Succiniclasticum* (Figure 2J) was significantly more abundant in the LP-L and LP-H groups than in the LP-M group (*p* < 0.05).

*Ruminococcaceae UCG-014*, *Saccharoferments*, and *uncultured rumen bacteria* were the major microbial communities in LP-H, LP-M, and LP-L groups, respectively (Figure 3A). The abundance comparison chart of functional items with statistical variations between groups in distinct groups and the LDA value distribution bar chart. LDA value larger than 2 indicated functional items with statistical differences across groups (Figure 3B). Carbohydrate metabolism, molecules and transport, and membrane transport are the most important items in the LP-H, LP-M, and LP-L groups, respectively.

### 3.5. Non-Targeted Metabolomics Analysis of Rumen Microbiota

#### 3.5.1. Univariate Statistical Analysis

Differential analysis was applied to all metabolites with positive and negative ion modes (including unknown metabolites) based on univariate analysis. Volcano plots were used to display differences in the content of metabolites (with FC > 1.5, FC < 0.67, and *p* value < 0.05). For the positive ion modes, 164 (LP-H vs. LP-M), 295 (LP-H and LP-L), and 333 (LP-M and LP-L) metabolites were up-regulation, while 194 (LP-H vs. LP-M), 136 (LP-H and LP-L), and 160 (LP-M and LP-L) metabolites were down-regulation (Figure 4A). For the negative ion modes, 145 (LP-H vs. LP-M), 307 (LP-H and LP-L), and 302 (LP-M and LP-L) metabolites were up-regulation, while 208 (LP-H vs. LP-M), 361 (LP-H and LP-L), and 239 (LP-M and LP-L) metabolites were down-regulation (Figure 4B).

#### 3.5.2. Principal Component Analysis

The separation of positive and negative ion modes was satisfactory (Figure 5A). The PC1 between the groups was 12%, and the PC2 was 9.6% in the positive ion mode PCA scatter plot. Negative ion mode PC1 was 15.4%, and PC2 was 11.7% in the negative ion mode PCA scatter plot. The three sets of samples were further analyzed using OPLS-DA to maximize intergroup separation. The Permutation test is used to test the model’s effectiveness and prevent overfitting of supervised models throughout the modeling phase. The OPLS-DA of LP-H and LP-M groups (R2Y = 0.956, Q2Y = 0.160), LP-H and LP-L groups (R2Y = 0.991, Q2Y = 0.238), and LP-M and LP-L groups (R2Y = 0.923, Q2Y = 0.304) were detected in positive ion mode (Figure 5B). OPLS-DA of LP-H and LP-M groups (R2Y = 0.956, Q2Y = 0.160), LP-H and LP-L groups (R2Y = 0.991, Q2Y = 0.238), and LP-M and LP-L groups (R2Y = 0.923, Q2Y = 0.304) were detected in negative ion mode (Figure 5C). The R2 and Q2 of the random model steadily declined as the permutation retention gradually dropped, suggesting that the original model did not exhibit overfitting and had strong robustness. Matching databases were used to perform standard analysis using the projection variable importance (VIP) from OPLS-DA to identify biologically relevant metabolites (VI*P* > 1.0, *p* < 0.05) (Appendix A). Among all the metabolites, the contents of phosphoric acid, pyrocatechol, hydrocinnamic acid, banzamide, l-gulono-1,4-lactone, cis-jasmone, Val-Asp-Arg, and tropinone levels were significantly higher in group LP-L. l-citrulline and purine metabolic levels were lower in the LP-L group.

#### 3.5.3. Metabolic Pathways of Differential Metabolites

The Kyoto Encyclopedia of Genes and Genomes (KEGG) enrichment analysis of the different metabolites is shown in Figure 6A–C. Glutathione metabolism, glucagon signaling, amino acid biosynthesis, phenoline, tyrosine, and tryptophan production, glycoxylate and dicarboxylate metabolism, and carbon metabolism were the main metabolic processes that differed between the LP-L and LP-H groups. The primary metabolic pathways that differed between the LP-L and LP-M groups included bile secretion and the biosynthesis of unsaturated fatty acids. Carbon metabolism, pentose phase pathway, biosynthesis of unsaturated fatty acids, and glycolysis/gluconeogenesis were the main metabolic pathways that differed between the LP-H and LP-M groups (Appendix A).

### 3.6. Microbiome–Metabolome–Phenotypic Index Joint Analysis 

Joint microbiomics–metabolomics–phenotypic index analysis and Spearman’s correlation coefficient were used to analyze the correlation between the abundance of different microorganisms and the different metabolites. The results showed that at the phylum level (Figure 7A), the abundance of *Firmicutes*, *Proteobacteria*, and *Acidobacteria* was associated with the content of several metabolites, including pyrocatechol, phosphoric acid, hydrocinnamic acid, banzamide, L-gulono-1,4-lactone, cis-jasmone, and Val-Asp-Arg. At the genus level (Figure 7B), the abundance of uncultured bacteria was negatively correlated with all differential metabolites, while the abundance of *Prevotella 1* and *Succiniclasticum* was positively correlated with the differential metabolites phosphoric acid and hydrocinnamic acid, the abundance of *Rikenellaceae RC9 gut group* and *Succiniclasticum* was positively correlated with the differential metabolites pyrocatechol, banzamide, L-gulono-1,4-lactone, cis-jasmone, and Val-Asp-Arg. Microbial–phenotypic correlation analysis (Figure 7C) revealed that the abundance of *Acidobacteria* was positively correlated with several antioxidant indicators, including GSH-Px, T-AOC, and SOD, and the activities of digestive enzymes α-amylase and cellulase. Differential metabolite–phenotype correlation analysis (Figure 7D) revealed that the content of banzamide, cis-jasmone, and Val-Asp-Arg was positively correlated with phenotypic indices, including T-AOC, SOD, and cellulase. The content of phosphoric acid was positively correlated with the activities of cellulase and lipase.

## 4. Discussion

T-AOC level is a measure of the body’s overall antioxidant ability. The primary antioxidant enzymes are CAT, SOD, and GSH Px. Lipid peroxidation levels in the body can be assessed by measuring the MDA level [22]. Liang et al. (2019). found that adding *N*-acetyl-l-methionin to a diet reduces lipid peroxidation and increases hepatic protein synthesis in in mid-lactating dairy cows [23]. It was found that methionine and lysine supplementation affects the antioxidant capacity in sheep, while methionine supplementation can reduce MDA contents and increase the levels of SOD, CAT, phosphoglycolate phosphatase, and glutathione S-transferase [9]. The study showed that a diet supplemented with lysine/methionine at a ratio of 1:1 increased the T-AOC, GSH Px, and SOD levels. The MDA content was lower, which is consistent with the above studies. KEGG analysis showed that Lysine and methionine exert the above effect by activating the Keap1-Nrf2/ARE signaling pathway. The Keap1-Nrf2/ARE signaling pathway ameliorates oxidative stress in animals [24]. Amino acids are the building blocks for protein synthesis, and Nrf2 is an endogenous antioxidant and a protein involved in the formation of red blood cells and platelets. Our results indicated that the 1:1 lysine-to-methionine ratio is optimal in Tibetan sheep. Nrf2 induces amino acid metabolism and transport through the ARE pathway to regulate the metabolism of many reactive oxygen species in the body, thereby activating the Keap1-Nrf2/ARE signaling pathway.

In animals, digestion and absorption of food primarily occur in the rumen, and an animal’s capacity for these processes can be assessed by monitoring the activity of the digestive enzymes [25]. Microorganisms in the rumen of ruminants secrete several digestive enzymes, and the activity of these enzymes is mostly determined by the kind, amount, and developmental stage of these microbiota. Digestive enzymes break down complex nutrients into simpler molecules that can be utilized by animals [26]. The proportion of amino acids in the diet impacts the enzymatic activity of rumen microorganisms. Research indicates that supplementing sheep’s food with 1% isoacids (i-C4, i-C5, C5, and phenylacetic acid) stimulates the growth of bacteria that break down cellulose and increases the secretion of cellulase in their rumen [27]. Cellulase breaks down cellulose into glucose or oligosaccharides, which animals then use for growth. Garnot et al. [28] found dietary supplementation with casein increases the secretion of gastric protease from the basal gland cells in the abomasum. The present study revealed different Lys/Met ratios in the diet affect several activities in the rumen. They also substantially impact the activities of trypsin, lipase, chymotrypsin, and α-amylase. The effect was lower in the LP-L group than in the LP-M and LP-H groups. The abundance and activity of bacteria in the rumen that break down fat and carbohydrates are unaffected by variations in the amino acid proportions in the diet. The cellulase activity was significantly higher in the LP-L group than in the LP-H group. Specifically, compared with the 2:1 group, the increase in the LP-L group went up by 73.14, but the difference was insignificant. This may be due to the fact that the rumen is the main site for the digestion of crude fiber in feed, whose degradation rate is about 55–75% [29]. Interestingly, with the addition of fiber-degrading bacteria, fiber in the rumen is digested better by various enzymes secreted by rumen microorganisms, including cellulase and cellulase complexes. Therefore, Lys/Met ratio at 1:1 in the diet could promote the secretion of digestive enzymes in the rumen.

Rumen pH is a crucial parameter that regulates the rate of rumen fermentation and the breakdown of nutrients. It is also a crucial marker for microbial metabolisms of organic matter, absorption, and release. Rumen fluid has a pH that ranges from 5.5 to 7.5. Very low pH can cause metabolic disorders and other damage in the rumen pH [30]. Tamura et al. [31] found that dietary supplementation with lysine and methionine had no effect on Holstein cows’ mik pH. In the present study, no significant pH difference was observed between LP-L, LP-M, and LP-H groups. In the present study, the pH of the rumen fluid was within the normal range of 6.58–6.79, suggesting that varying the amounts of amino acids in the food does not negatively impact the rumen pH of Tibetan sheep. The primary byproduct of rumen microbial fermentation is VFA, whose content can be used to measure variations in rumen fermentation patterns, supplies energy to the animals, supports rumen development, stimulates the growth of intestinal wall cells, and reach the systemic circulation through capillaries [32,33]. Research has demonstrated that tryptophan supplementation, an amino acid, has an insignificant effect on microbial fermentation in vitro [34], which differs from our findings. Here, acetic acid content in the rumen was significantly higher in the 1:1 and 2:1 groups than in the 3:1 group. Acetic acid is one of the main products of carbohydrate metabolism in the rumen, which affects protein metabolism and other functions. The present study revealed that a high proportion of amino acids in the diet does not necessarily offer beneficial outcomes. Very high amino acid supplementation in feeds hinders the digestion of fibers. In the present study, the ammonia nitrogen content in the 1:1 group was significantly lower than in the LP-M and LP-H groups by 13.2% and 15.6%, respectively. A dietary Lys/Met ratio at 1:1 reduced the production of harmful gases and maximized the utilization of protein resources.

The diversity of rumen microbiota in ruminants affects the digestion and metabolism of nutrients [35]. The α diversity indices reflect the richness and diversity of species in the rumen [36]. The larger the Shannon index, the higher the species diversity of microbial communities [37]. In this experiment, no significant difference was observed in the Shannon index between the LP-L, LP-M, and LP-H groups, indicating that different Lys/Met ratios had no effect on the diversity of rumen microbiota in the diet with low protein levels. The Ace and Chao1 indices in this experiment were significantly higher in the LP-L group than in the LP-M and LP-H groups. One possible explanation is that the protein in the diet was broken into peptides, amino acids, and ammonia, which occurs in the rumen. These products of protein breakdown supply energy to the rumen microbiota. Appropriate amount and diversity of rumen microbiota is essential for the development of several bacterial species. The maximum utilization rate occurred at a ratio 1:1 Lys/Met ratio.

Our study revealed that *Firmicutes*, *Bacteroidetes*, and *Proteobacteria* are the dominant rumen microbiota in Tibetan sheep, consistent with Minseok Kim et al. [38] findings. *Firmicutes*, members of the Gram-positive bacterial family, participate in the breakdown and modification of organic matter, which enhances the metabolism and absorption in ruminants [39]. *Bacteroidetes* break down non-fibrous materials, which promote the digestion and absorption of proteins and non-cellulose polysaccharides. Non-fibrous materials are broken down to propionic acid, which is used to synthesize bacterial proteins and promote the absorption of amino acids [40]. The *Proteobacteria* phylum is primarily involved in the digestion of soluble carbohydrates and the development of biofilms. It also produces LPS, and its flagellar proteins induce inflammatory reactions, which can be harmful to animal bodies [41]. Using in vitro rumen simulation tests, An et al. (2023) found that adding leucine to feeds regulates rumen fermentation and increases the relative abundance of *Firmicutes* in vitro [42]. According to Liu et al. (2022), feeding yaks with a low-protein diet enhanced the abundance of *Bacteroidetes* to *Firmicutes* (F:B) ratio [43]. In the present study, *Bacteroidetes* was more dominant in the LP-L and LP-H groups than in the LP-M group, and this could be caused by rumen fermentation that yields propionic acid. Ruminants derive their energy from gluconeogenesis of propionic acid in the liver. Consequently, the growth of Tibetan sheep was best in the LP-L and LP-H groups. The abundance of *Firmicutes* was significantly higher in the LP-L and LP-M groups than in the LP-H group. Meanwhile, the abundance of *Proteobacteria* was significantly higher in the LP-M group than in the LP-L and LP-H groups. At a Lys/Met ratio of 1:1, there was a significant and positive change in the abundance of rumen microbiota in Tibetan sheep, which improved the digestion and absorption of nutrients and reduced the abundance of harmful bacteria, making the rumen environment healthier. At the phylum level, *Acidobacteria* was dominant in the LP-L group. *Acidobacteria* participate in the nitrite decomposition, and active protein transport [44]. However, the specific mechanism of action in Tibetan sheep needs further investigation.

*Prevotella 1*, the *Rikenellaceae RC9 gut group*, and *uncultured rumen bacteria* were the dominant bacteria genera in the rumen of Tibetan sheep. With the exception of *Prevotella*, the abundance of uncultivated rumen bacteria is equal to or higher than that of known bacterial species or genera significant to rumen function [38]. Similar to certain completely defined bacteria, the abundance and relationships of uncultivated rumen bacteria with recognized taxa imply that some bacteria may be crucial to rumen function [45]. *Quinella* is a characteristic genus of rumen bacteria in uncultured rumen bacteria [46]. *Quinella* breaks down to produce propionic acid, lowering methane emissions in sheep. The mechanism by which this bacterium decreases methane emissions in sheep is not well understood [47]. In the present study, we found that a low-protein diet reduces methane emissions. In addition, the abundance of uncultured rumen bacteria was higher in the LP-H group than in the LP-M group. Kittelmann et al. [48] revealed that sheep exhibiting a rise in the *Quinila* bacterial population had a higher relative rumen propionic acid concentration, consistent with our findings. *Prestotella* plays a key role in the breakdown of cellulose-based plant fibers. *Prestotella* can also recognize, bind, and break down different polysaccharides and proteins using the products of polysaccharide utilization sites (PULs) genes [49]. The abundance of *Prevotella* was higher in the LP-H and LP-M groups than in the LP-L group. The availability of free amino acids by microbes may have increased, increasing the microbial abundance, suggesting that an increase in microbial abundance parallels microbial protein synthesis. Within the *Rikenellaceae* family, the *Rikenellaceae RC9 gut group* is the main bacteria that participates in the secondary breakdown of structural carbohydrates [50]. Due to the high concentration of *Rikenellaceae RC9* in the rumen of grazing sheep, they can stimulate the deoxycholic acid in lamb meat and the deposition of linolenic and glycocholic acids, which can regulate lipid glucose metabolism and the subsequent energy production in the host’s body [51]. Interestingly, in the present study, the abundance of *Saccharoferments* was substantially higher in the LP-L group than in the LP-H and LP-M groups. The degradation of polysaccharides was higher in the LP-L group since *saccharoferments* are non-cultivated phylogenetic bacteria that can break down polysaccharides into ethanol, acetate, hydrogen, and carbon dioxide. Further analyses in the present study revealed that the abundance of *Succiniclasticum* increased with a decrease in Lys/Met ratio.

Rumen and host health are affected by altered concentrations of various rumen metabolites. In the present study, PCA and OPLS-DA analyses revealed a significant separation of samples among the three groups, demonstrating that different Lys/Met ratios significantly affected rumen metabolism. In this study, the L-citrulline content of the LP-H and LP-M groups was significantly higher than that of the LP-L group. A separate study showed that l-citrulline effectively reduced body temperature, increased heat tolerance, and improved animal stress tolerance [52]. l-citrulline improves vascular function and exercise capacity by increasing l-arginine bioavailability and nitric oxide synthesis [53]. This suggests that as the Lys/Met ratio increases, the higher l-citrulline content produced reduces heat stress in Tibetan sheep and improves meat quality by increasing the production of the rumen metabolite l-citrulline, resulting in improved athletic performance in those who consume it. However, the exact mechanism by which this happens remains elusive. Purine level is an indicator of protein supply in animals [54]. The LP-H and LP-M groups were significantly higher than the LP-L group, and the increase positively correlated with an increase in Lys/Met ratio. Cis-jasmone is a volatile compound in plants that increases plant resistance, produces defense-related volatile chemicals, and can be used as a flavoring agent in food [55]. Our results showed that cis-jasmone content was significantly higher in the LP-L group than in the LP-H and LP-M groups, which might have enhanced the flavor of Tibetan lamb meat. However, whether this is true remains to be investigated. Tsiftsoglou et al. (2023) showed that cis-jasmone, present in essential oils of plants, possesses antioxidant properties [56]. The antioxidant capacities were significantly higher in the LP-L group than in the LP-H and LP-M groups, suggesting that as the Lys/Met ratio decreases, the metabolite cis-jasmone increases, improving the antioxidant capacity of the organism. The metabolite Val-Asp-Arg content was significantly higher in the LP-L group than in the LP-H and LP-M groups, suggesting that more free amino acids are produced after protein synthesis and conversion at a Lys/Met ratio of 1:1, and they are very beneficial to the rumen microflora as well as a source of food for the health and growth of the animal. Tropinone can be converted into a reductase enzyme [57], which was secreted at the highest level in the LP-L group in the present study, and this may explain the high digestive enzyme activity in the rumen at a Lys/Met ratio of 1:1. Derivatives of banzamide play important roles in pharmacology and pathology [58], and it has been shown that derivatives of banzamide can lower blood glucose in humans, making them good candidates for treating diabetes [59]. The glucagon signaling pathway, a metabolic pathway of banzamide synthesis/breakdown, was a major metabolic pathway in the 1:1 group. The content of metabolite pyrocatechol increased with a decrease in Lys/Met ratio, and pyrocatechol can inhibit LPS-induced inflammation by inhibiting NF-κB but activating Nrf2 signaling [60]. Pyrocatechol in KEGG is mainly involved in the metabolism of benzoate degradation and the degradation of aromatic compounds. Benzoate degradation enhances animal health. The content of the metabolite hydrocinnamic acid was higher in the Lys/Met ratio of the LP-L group. Hydrocinnamic acid promotes the intestinal epithelial barrier function in the gut microbiota of crossbred pigs through the AHR signaling pathway [61]. The degradation of hydrocinnamic acid and aromatic compounds occurs in the same pathway. Interestingly, the content of phosphoric acid was higher in the LP-L group in the present study. Phosphoric acid is mainly involved in oxidative phosphorylation, a process that drives ATP production and energy release from the catabolism of amino acids and others. The concentration of succinic acid coenzymes in the tricarboxylic acid cycle increases with the abundance of *Succiniclasticum* flora.

## 5. Conclusions

Our results showed that a Lys/Met ratio of 1:1 increased the antioxidant capacity and the activities of digestive enzymes, as well as reduced the ammonia nitrogen in Tibetan sheep. In addition, 16S rDNA sequencing revealed that a Lys/Met ratio of 1:1 significantly increased the microbial diversity in the rumen. The non-target metabolomics analysis revealed that cis-jasmone and Val-Asp-Arg could be involved in the antioxidant capacity and digestive enzyme activity in Tibetan sheep. Phosphoric acid, one of the metabolic products, increased cellulase activity by regulating the abundance of *Succiniclasticum* through the oxidative phosphorylation pathway. However, further research is required to accurately define the optimal Lys/Met ratio in Tibetan sheep diets at varying CP levels and to better quantify the mechanisms impacting the Lys/Met requirement.

## Figures and Tables

**Figure 1 animals-14-01533-f001:**
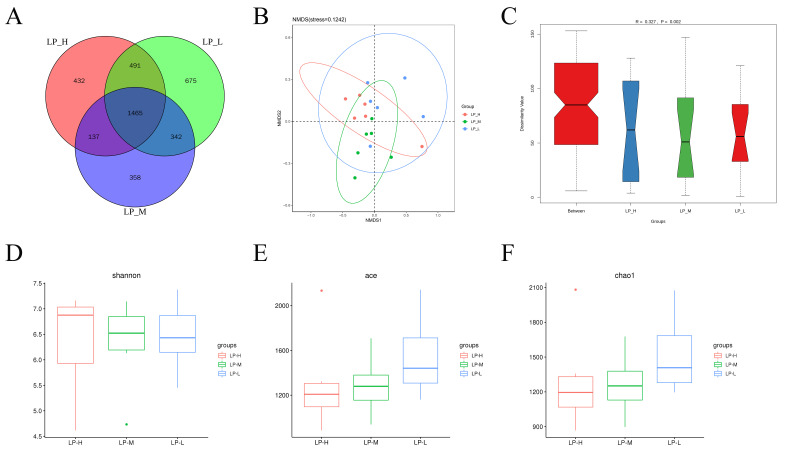
Rumen microbial diversity among the three groups. Venn diagram showing the number of common or unique OTUs (**A**). Principal component analysis (PCA) by the weighted Unifrac of beta diversity (**B**). Anosim analysis (**C**). Shannon index box plot (**D**). Ace index box plot (**E**). Chao1 index box plot (**F**). LP-H, LP diet supplemented with the Lys/Met ratio at 3. LP-M, LP diet supplemented with the Lys/Met ratio at 2. LP-L, LP diet supplemented with the Lys/Met ratio at 1.

**Figure 2 animals-14-01533-f002:**
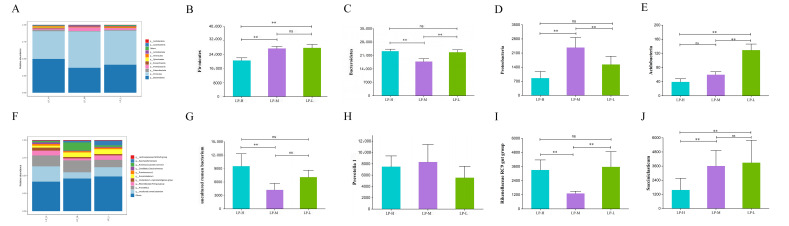
Composition and differences of the highly abundant microbial community at different Lys/Met ratios in the rumen. Percentage composition of the predominant phyla (**A**). The relative abundance of Firmicutes (**B**), Bacteroidetes (**C**), Proteobacteria (**D**), and Acidobacteria (**E**). Percentage composition of the predominant genus (**F**). The relative abundance of uncultured rumen bacterium (**G**), Prevotella 1 (**H**), Rikenellaceae RC9 gut group (**I**), and Succiniclasticum (**J**). ns represent no significant difference. ** *p* < 0.05. LP-H, LP diet supplemented with the Lys/Met ratio at 3. LP-M, LP diet supplemented with the Lys/Met ratio at 2. LP-L, LP diet supplemented with the Lys/Met ratio at 1.

**Figure 3 animals-14-01533-f003:**
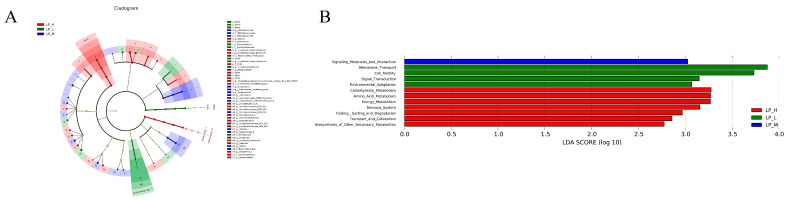
The circles radiating from the inside out in the cladistic diagram represent taxonomic levels from phylum to genus (or species). The red, blue, and green areas represent different groups, and the red, blue, and green nodes in the branches represent microbial groups that play an important role in each group, while the yellow nodes represent microbial groups that play no important role in any of the three groups (**A**). Significantly different bacterial taxa identified by the linear discriminant analysis effect size (**B**). LP-H, LP diet supplemented with the Lys/Met ratio at 3. LP-M, LP diet supplemented with the Lys/Met ratio at 2. LP-L, LP diet supplemented with the Lys/Met ratio at 1.

**Figure 4 animals-14-01533-f004:**
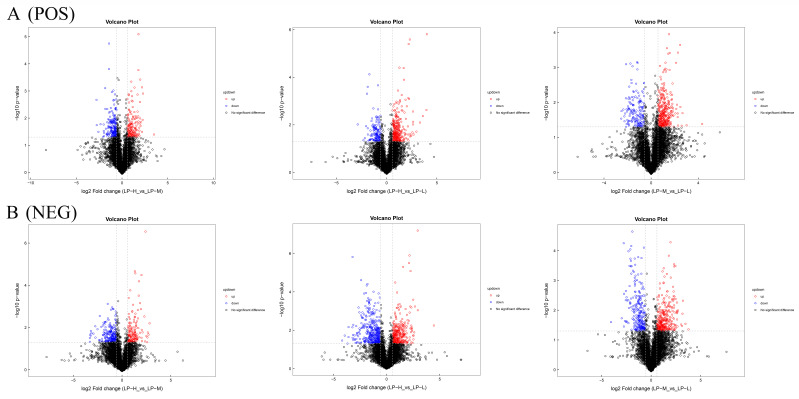
Volcano diagram of three groups of metabolites. Positive ion mode (**A**). Negative ion mode (**B**). LP-H, LP diet supplemented with the Lys/Met ratio at 3. LP-M, LP diet supplemented with the Lys/Met ratio at 2. LP-L, LP diet supplemented with the Lys/Met ratio at 1.

**Figure 5 animals-14-01533-f005:**
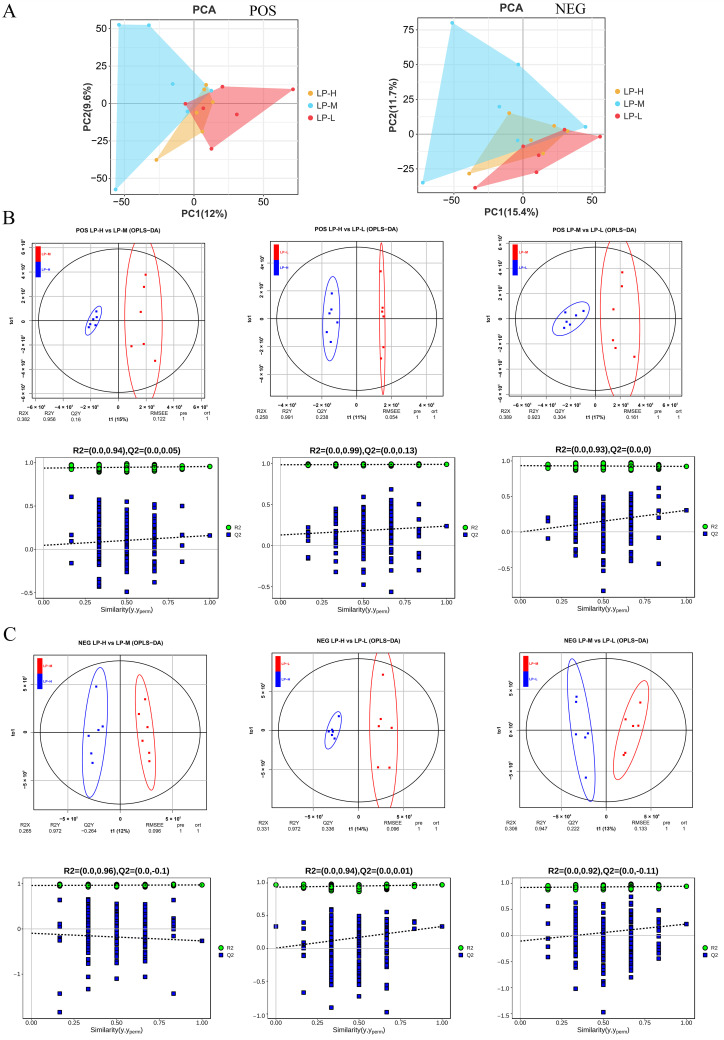
Principal component analysis (PCA) model score scatter plot, orthogonal partial least-squares discriminant analysis (OPLS-DA) model, and permutation test of rumen metabolic profiling. Principal component analysis (PCA) in positive and negative ion modes (**A**). Three sets of OPLS-DA diagrams and three sets of permutation test diagrams for positive ion mode (**B**). Three sets of OPLS-DA diagrams for negative ion mode and three sets of permutation test diagrams (**C**). LP-H, LP diet supplemented with the Lys/Met ratio at 3. LP-M, LP diet supplemented with the Lys/Met ratio at 2. LP-L, LP diet supplemented with the Lys/Met ratio at 1.

**Figure 6 animals-14-01533-f006:**
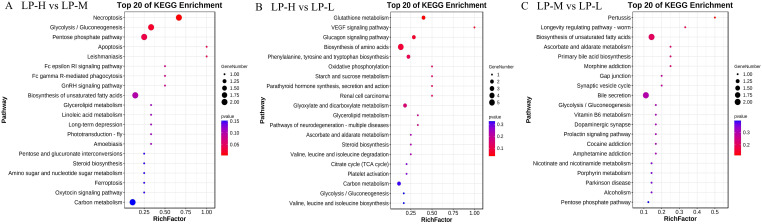
Kyoto Encyclopedia of Genes and Genomes (KEGG) analysis between treatment groups. LP-H vs. LP-M (**A**). LP-H vs. LP-L (**B**). LP-M vs. LP-L (**C**). LP-H, LP diet supplemented with the Lys/Met ratio at 3. LP-M, LP diet supplemented with the Lys/Met ratio at 2. LP-L, LP diet supplemented with the Lys/Met ratio at 1.

**Figure 7 animals-14-01533-f007:**
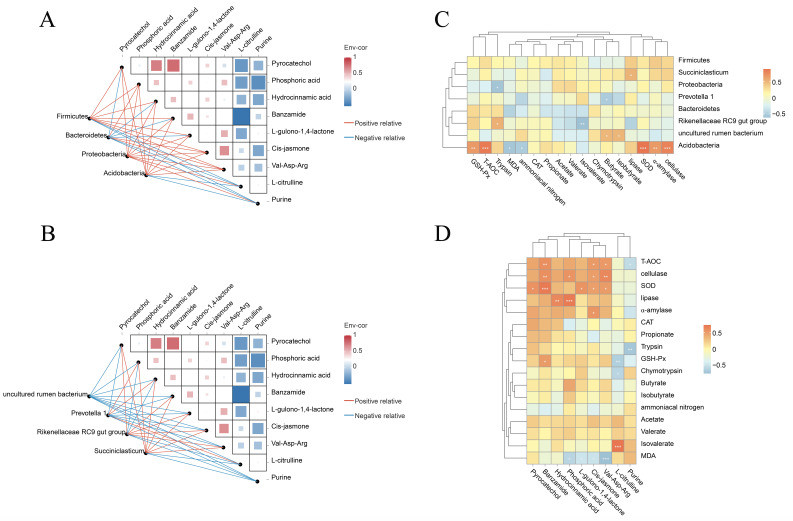
Microbiome–metabolome network heat map (**A**,**B**). Heat map of the correlation between microbiome and phenotype (**C**). Heat map of correlation between metabolomics and phenotype (**D**). * *p*-value < 0.05, ** *p*-value < 0.01, and *** *p*-value < 0.001.

**Table 1 animals-14-01533-t001:** Dietary concentrate composition and nutrient levels (dry matter basis).

Items	LP-L	LP-M	LP-H
Ingredient (%)			
Oat hay	15.000	15.000	15.000
Oat silage	15.000	15.000	15.000
Corn	36.533	37.100	37.100
Wheat	7.700	7.700	7.700
Soybean meal	0.700	0.700	0.700
Rapeseed meal	7.000	7.000	7.000
Cottonseed meal	0.700	0.700	0.700
Maize germ meal	0.700	0.700	0.700
Palm meal	11.200	11.200	11.200
NaCl	0.350	0.350	0.602
Limestone	0.350	0.441	0.700
Baking soda	0.070	0	0.070
Premix ^(1)^	2.940	2.940	2.940
Lys	1.386	0.931	0.483
Met	0.371	0.238	0.105
Total	100.000	100.000	100.000
Nutrient levels			
DE (MJ·kg^−1^) ^(2)^	10.760	10.840	10.840
Crude protein	9.940	9.980	9.980
Ether extract	2.850	2.870	2.870
Crude fiber	22.470	22.61	22.610
Neutral detergent fiber	33.720	33.77	33.770
Acid detergent fiber	23.370	23.39	23.390
Ca	0.421	0.424	0.424
P	0.171	0.172	0.172

^(1)^ Provided per kilogram of diets: Cu 15 mg, Fe 55 mg, Zn 25 mg, Mn40 mg, Se 0.30 mg, I 0.5 mg, Co 0.20 mg, VA 20,000 IU, VD 4000 IU, VE 40 IU. ^(2)^ Digestive energy is the calculated value, while the rest are measured values. LP-H, LP diet supplemented with the Lys/Met ratio at 3. LP-M, LP diet supplemented with the Lys/Met ratio at 2. LP-L, LP diet supplemented with the Lys/Met ratio at 1.

**Table 2 animals-14-01533-t002:** Determination of antioxidant index.

Items	LP-H	LP-M	LP-L	*p*-Value
T-AOC	7.95 ± 0.43 ^b^	8.36 ± 0.55 ^b^	10.58 ± 0.28 ^a^	0.004
CAT	124.14 ± 8.32	123.57 ± 7.81	126.94 ± 4.72	0.938
GSH-Px	262.59 ± 21.30 ^b^	218.61 ± 2.24 ^b^	320.93 ± 19.94 ^a^	0.007
SOD	86.90 ± 10.61 ^b^	106.01 ± 10.21 ^b^	156.73 ± 6.98 ^a^	0.001
MDA	8.64 ± 0.16	8.52 ± 0.14	7.76 ± 0.93	0.514

^a,b^ Means with different superscripts in the same row are significantly different (*p* < 0.05). Data are presented as mean ± SEM. LP-H, LP diet supplemented with the Lys/Met ratio at 3. LP-M, LP diet supplemented with the Lys/Met ratio at 2. LP-L, LP diet supplemented with the Lys/Met ratio at 1. T-AOC, total antioxidant capacity. CAT, catalase. GSH-Px, glutathione peroxidase. SOD, superoxide dismutase. MDA, malondialdehyde.

**Table 3 animals-14-01533-t003:** Determination of digestive enzyme activity.

Items	LP-H	LP-M	LP-L	*p*-Value
α-amylase	198.80 ± 1.79	213.42 ± 7.03	217.22 ± 3.30	0.065
Chymotrypsin	182.12 ± 1.09	163.03 ± 23.79	174.55 ± 10.92	0.685
Cellulase	135.32 ± 2.94 ^b^	242.27 ± 12.06 ^a,b^	315.41 ± 50.71 ^a^	0.027
Trypsin	509.61 ± 21.65	492.75 ± 32.85	529.22 ± 34.24	0.707
Lipase	509.61 ± 21.65	560.47 ± 49.76	568.47 ± 56.77	0.617

^a,b^ Means with different superscripts in the same row are significantly different (*p* < 0.05). Data are presented as mean ± SEM. LP-H, LP diet supplemented with the Lys/Met ratio at 3. LP-M, LP diet supplemented with the Lys/Met ratio at 2. LP-L, LP diet supplemented with the Lys/Met ratio at 1.

**Table 4 animals-14-01533-t004:** Fermentation parameter determination.

Items	LP-H	LP-M	LP-L	*p*-Value
Acetate/(mmol/mL)	59.23 ± 8.62 ^b^	87.43 ± 4.25 ^a^	74.20 ± 3.06 ^a^	0.039
Propionate/(mmol/mL)	20.45 ± 1.53	21.60 ± 1.13	22.25 ± 1.62	0.794
Butyrate/(mmol/mL)	9.63 ± 1.51	10.07 ± 3.41	10.39 ± 0.41	0.934
Isobutyrate(mmol/mL)	2.01 ± 0.13	1.89 ± 0.16	1.97 ± 0.11	0.840
Valerate/(mmol/mL)	1.63 ± 0.14	2.53 ± 0.28	1.97 ± 0.02	0.054
Isovalerate/(mmol/mL)	3.53 ± 0.24	4.31 ± 0.28	3.51 ± 0.23	0.109
Ammonia nitrogen (mg/L)	256.70 ± 6.52 ^a^	242.97 ± 6.48 ^a^	202.65 ± 11.17 ^b^	0.042
pH	6.58 ± 0.16	6.65 ± 0.07	6.79 ± 0.03	0.306

^a,b^ Means with different superscripts in the same row are significantly different (*p* < 0.05). Data are presented as mean ± SEM. LP-H, LP diet supplemented with the Lys/Met ratio at 3. LP-M, LP diet supplemented with the Lys/Met ratio at 2. LP-L, LP diet supplemented with the Lys/Met ratio at 1.

## Data Availability

The datasets presented in this study can be found in online repositories. The names of the repository/repositories and accession number(s) can be found below: NCBI SRA (accession: PRJNA1095164). MetaboLights (MTBLS9939 https://www.ebi.ac.uk/metabolights/editor/study/MTBLS9939, accessed on 30 April 2024).

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
