# Peer review of "Changes in Rumen Microbiology and Metabolism of Tibetan Sheep with Different Lys/Met Ratios in Low-Protein Diets"

_animals, 2024, doi:10.3390/ani14111533_

Round 1
Reviewer 1 Report
Comments and Suggestions for Authors
-Line 63: Lowering the amount of protein in the diet can lower nitrogen emissions and feed costs the connection between increasing the amount of amino acids in feeds to meet animal growth requirements and balanced amino 65 acids in the body is not clear.
Please correct the highlighted text
-Please include more literature on amino acids in ruminants, references to monogastrics could possibly be omitted.
-Please explain in text in detail what were the sources of protein in the rations (information are only available in Table 1)
- No reference on oxidative status and antioxidants is done in the introduction. Please correct.
Comments on the Quality of English Language
Quite a few spelling mistakes are encountered throughout the text, as well as some grammar ones, probably due to overseeing.
Please review the text carefully and correct.
Author Response
Reviewer 1#
-Line 63: Lowering the amount of protein in the diet can lower nitrogen emissions and feed costs the connection between increasing the amount of amino acids in feeds to meet animal growth requirements and balanced amino acids in the body is not clear.
Please correct the highlighted text
Response: Thanks for your kind comment. This has been corrected accordingly.
-Please include more literature on amino acids in ruminants, references to monogastrics could possibly be omitted.
Response: Thanks for your kind comment. Several references of monogastrics were replaced with ruminants.
-Please explain in text in detail what were the sources of protein in the rations (information are only available in Table 1).
Response: Thanks for your kind comment. The information of protein sources was added in the Materials and Methods section.
- No reference on oxidative status and antioxidants is done in the introduction. Please correct.
Response: Thanks for your kind comment. The relevant research reports were supplied in the introduction section

Reviewer 2 Report
Comments and Suggestions for Authors
Overall this paper evaluated the changes in the ruminal microbiome of Tibetan sheep due to varying ratios of Lysine / Methionine. For this project, the authors collected a multitude of data, much of which had not really be done previously with these types of animals. I commend the authors for the amount of work put into the project and manuscript preparation. This work has the potential to open up new areas of research related to specific amino acid ratios within the diets ruminants. There are a few concerns, based upon the data and information presented which I feel need to be addressed.
General Comments
- The diets were formulated to be 12% protein and this is described as a "low-protein" diet, but how much below requirements is it and why was this level chosen for the overall protein level of the diet?
- Was the nutrient content of the forage tested and taken into account for the nutrient profile of the diet as a whole?
- The tables / figures should be able to stand alone if removed from the manuscript, so there should be sufficient description in them to fully understand what is presented without reading through the paper itself. Please adjust some of these tables accordingly. (For example, on Table 2, footnotes below the table to define the LP-H etc., as well T-AOC etc.)
- Double check references section as it appears that # 6 and # 33 are missing some author information.
Specific Comments
- Line 64: add a period after "feed costs" and capitalize the "t" in "the connection"
- Line 104: there appears to be missing words after "farmers in the"
- Line 122: Remove parenthesis around "Table 1"
- Line 124: Change "starved" to "held off feed"
- For Table 1: Is this DM% or AF %? "corn" should be capitalized; What is the difference between the 1% and 3.6% Premix (define in the footnote); How were the Lysine & Methionine supplied in the diet (rumen protected? dissolved in liquid?)
- Line 195 & 196: Based upon Table 3, it appears as though there is a statistical difference between LP-L & LP-H groups for cellulase. Which is correct?
- In Table 3: cellullase and lipase should be capitalized
- Line 202: Why was this described as "group 1:1" and "group 2:1" instead of "LP-L" & "LP-M" as it is throughout the bulk of the paper?
- Line 202 - 206: Please double check the data presented as this discussion does not agree with Table 4.
- In Table 4: Should "ammoniacal" be "ammonia" ? Also, it should be capitalized.
Comments on the Quality of English LanguageGenerally pretty good, just a few minor grammatical issues here and there.
Author Response
Reviewer 2#
Overall this paper evaluated the changes in the ruminal microbiome of Tibetan sheep due to varying ratios of Lysine / Methionine. For this project, the authors collected a multitude of data, much of which had not really be done previously with these types of animals. I commend the authors for the amount of work put into the project and manuscript preparation. This work has the potential to open up new areas of research related to specific amino acid ratios within the diets ruminants. There are a few concerns, based upon the data and information presented which I feel need to be addressed.
Response: Thank you for giving us a chance to revise the manuscript. We also thank the reviewers for constructive suggestions to help us improve the quality of the manuscript. We have made changes in the manuscript according to the reviewers’ comments and corresponding to your advice.
General Comments
- The diets were formulated to be 12% protein and this is described as a "low-protein" diet, but how much below requirements is it and why was this level chosen for the overall protein level of the diet?
Response: Thanks for your kind comment. Our feeding trial was conducted in Guinan County, Qinghai province of China (above 3,000 m altitude). The high altitude means that Tibetan sheep need more nutritional requirements for survival. Currently, normal dietary crude protein in the concentration is about 14-16% in the Tibetan sheep production. Recent nutritional guidelines have suggested that supplementation of amino acids in diets were the effective strategies for alleviating low-protein diets induced reduced growth performance in livestock. However, excessive reducing the dietary protein levels make it difficult to meet nutritional requirements for Tibetan sheep, resulting in growth retardation, hypoimmunity, and a higher mortality rate. Therefore, dietary CP content is decreased by 2% than the normal dietary crude protein in Tibetan sheep production.
- Was the nutrient content of the forage tested and taken into account for the nutrient profile of the diet as a whole?
Response: Thanks for your kind comment. The nutrient contents of the forage, including crude protein, crude protein, and crude fiber, etc, were measured. Additionally, the nutrient contents of diet (oat hay, oat silage and concentration) were calculated according to the proportion (dry matter basis).
- The tables / figures should be able to stand alone if removed from the manuscript, so there should be sufficient description in them to fully understand what is presented without reading through the paper itself. Please adjust some of these tables accordingly. (For example, on Table 2, footnotes below the table to define the LP-H etc., as well T-AOC etc.)
Response: Thanks for your kind comment. The details were provided according to the contents of tables/figures.
- Double check references section as it appears that # 6 and # 33 are missing some author information.
Response: Thanks for your kind comment. This has been corrected accordingly.
Specific Comments
- Line 64: add a period after "feed costs" and capitalize the "t" in "the connection"
Response: Thanks for your kind comment. This has been corrected accordingly.
- Line 104: there appears to be missing words after "farmers in the"
Response: Thanks for your kind comment. This sentence was revised.
- Line 122: Remove parenthesis around "Table 1"
Response: Thanks for your kind comment. The parenthesis around “Table 1” was removed.
- Line 124: Change "starved" to "held off feed"
Response: Thanks for your kind comment. The “starved” was changed to the “held off feed” in manuscript.
- For Table 1: Is this DM% or AF %? "corn" should be capitalized; What is the difference between the 1% and 3.6% Premix (define in the footnote); How were the Lysine & Methionine supplied in the diet (rumen protected? dissolved in liquid?)
Response: Thanks for your kind comment. The ingredients of diet were calculated by dry matter basis. The composition of Premix was added in the footnote. In this study, three treatments were fed different rumen-protected Lys/rumen-protected Met ratios. Both Lysine and Methionine were directly mixed with concentration.
Previous study have shown that reducing dietary CP with rumen-protected Met supplementation did not limit milk yield, milk composition or digestibility of nutrients, but could improve nitrogen utilization, synthesis of MCP and partially increase VFA production 2 h after feeding cows (Li et al., 2023). Similarly, the replacement of soybean meal (SBM) with Distillers dried grains with solubles (DDGS) with rumen-protected Lys + rumen-protected Met in diets with adequate metabolizable protein and bypass amino acids (lysine and methionine) could alter the rumen fermentation of Hu sheep (Chen et al., 2021). Therefore, we hypothesized that dietary Lys/Met ratios would change the rumen fermentation by regulate the microorganism and metabolism of Tibetan sheep.
Li, YX., Wei, JL., Dou, MY., Liu, S., Yan, BC., Li, CY., Khan, MZ., Zhang, YH., Xiao, JX. 2023. Effects of rumen-protected methionine supplementation on production performance, apparent digestibility, blood parameters, and ruminal fermentation of lactating Holstein dairy cows. Frontiers in veterinary science, 9, 981757. DOI:10.3389/fvets.2022.981757.
Chen, J., Niu, XL., Li, F., Li, FD., Guo, L. 2021. Replacing soybean Meal with distillers dried grains with solubles plus rumen-protected Lysine and Methionine: Effects on growth performance, nutrients digestion, rumen fermentation, and serum parameters in Hu Sheep. Animals, 11, 2428. DOI:10.3390/ani11082428.
- Line 195 & 196: Based upon Table 3, it appears as though there is a statistical difference between LP-L & LP-H groups for cellulase. Which is correct?
Response: Thanks for your kind comment. I am sorry for those mistakes. The descriptions for Tables were rewritten.
- In Table 3: cellullase and lipase should be capitalized
Response: Thanks for your kind comment. This has been corrected accordingly.
- Line 202: Why was this described as "group 1:1" and "group 2:1" instead of "LP-L" & "LP-M" as it is throughout the bulk of the paper?
Response: Thanks for your kind comment. The “group a:b” were replaced with “LP-X” in whole manuscript.
- Line 202 - 206: Please double check the data presented as this discussion does not agree with Table 4.
Response: Thanks for your kind comment. I am sorry for those mistakes. The descriptions for Tables were rewritten.
- In Table 4: Should "ammoniacal" be "ammonia" ? Also, it should be capitalized
Response: Thanks for your kind comment. The “ammoniacal” was changed to the “ammonia” in whole manuscript.

Reviewer 3 Report
Comments and Suggestions for Authors
Reviewer’s comments on the manuscript by Zhang et al. entitled: Changes in rumen microbiology and metabolism of Tibetan sheep with different Lys/Met ratios in low protein diets.
Manuscript ID: animals-2996531.
May, 2024.
This manuscript is very interesting, but the material and methods need to revise.
Specific Comments that are listed for the different section below:
L118-119: how many sheep in one pen? Please clarify in the manuscript.
L120-121: 70% concentrate??? does the authors not consider the cost of feeding?
L121-122: please explain why methionine and lysine ratios were 1:1 for LP-L, 2:1 for LP-M, and 3:1 for the LP-H group?? How to get these ratios???
Table 1:
How about the nutrient requirement for Tibetan sheep? Because the manuscript was ……..in low protein diets. However, I do not think the 12% CP is low protein, the authors should be provide a detailed explanation. I never seen any content for related nutrient requirement information.
In addition, please added the detail procedures for analysis chemical composition of diet.
L131-139: Are you sure you can detect antioxidant activity in ruminal fluid? Please add the detail steps for how to prepare ruminal fluid to detect these parameters. The same as α -amylase activity.
L147-148: please added the detail information for VFA determine. Such as, GC condition, etc.
L180-182: Have all data been tested for normality or not? And how to test?
L195: Some descriptions are incorrect, such as, No significant difference (P<0.05) was observed, please check carefully in the Result section.
L572-583: please add limitations for this manuscript in the Conclusion setion.
Author Response
Reviewer 3#
This manuscript is very interesting, but the material and methods need to revise.
Response: Thank you for giving us a chance to revise the manuscript. We also thank the reviewers for constructive suggestions to help us improve the quality of the manuscript. We have made changes in the manuscript according to the reviewers’ comments and corresponding to your advice.
Specific Comments that are listed for the different section below:
L118-119: how many sheep in one pen? Please clarify in the manuscript.
Response: Thanks for your kind comment. The rams were randomly split into three groups, each containing thirty rams, and housed in separate pens (30 Tibetan sheep per pen).
L120-121: 70% concentrate??? does the authors not consider the cost of feeding?
Response: Thanks for your kind comment. High concentration in diets increased the the cost of feeding. On the contrary, high concentration in diets shorted the breeding cycle and improved breeding efficiency through meting the nutritional requirements of Tibetan sheep. Previously, our results showed that 70% concentration in diet promoted the fermentation characteristics, ruminal bacterial community composition and meat quality in Tibetan sheep. Here, maintaining dietary 70% concentration (dry matter basis), the objective of the present study was to examine the effect of feeding with different ratio of Lys:Met on microbiome and metabolome of rumen. It may provide a theoretical basis for the low protein diet of feeding and management of Tibetan sheep.
L121-122: please explain why methionine and lysine ratios were 1:1 for LP-L, 2:1 for LP-M, and 3:1 for the LP-H group?? How to get these ratios???
Response: Thanks for your kind comment. At present, researches in ruminant mainly focus on the lysine (or methionine) alone or in combination. There is little comparative research on the effects of different ratio of Lys/Met on rumen microflora and metabolites in sheep. Previous research has shown that decreasing the dietary Lys/Met ratio affected the placental angiogenesis and piglet development (Peng et al. 2020). And the dietary Lys/Met ratios were from 2.1 to 3.7.
Therefore, Tibetan sheep were randomly divided into 3 treatments, which were supplemented with Lys/Met ratio at 1, 2, and 3 in the basal diet, respectively.
Peng, J., Xia, M., Xiong, J., Cui, CB., Huang, NN., Zhou, YF., Wei, HK., Peng, J. 2020. Effect of Sows Gestational Methionine/Lysine Ratio on Maternal and Placental Hydrogen Sulfide Production. Animals, 10, 251. DOI:10.3390/ani10020251.
Table 1:
How about the nutrient requirement for Tibetan sheep? Because the manuscript was ……..in low protein diets. However, I do not think the 12% CP is low protein, the authors should be provide a detailed explanation. I never seen any content for related nutrient requirement information.
In addition, please added the detail procedures for analysis chemical composition of diet.
Response: Thanks for your kind comment. In this study, the concentration was formulated to be 12% protein, not the diet. The nutrient contents of diet were re-calculated according to the proportion (15% oat hay, 15% oat silage and 70% concentration). Actually, the dietary crude protein level is about 10% in our experiment.
Our feeding trial was conducted in Guinan County, Qinghai province of China (above 3,000 m altitude). The high altitude means that Tibetan sheep need more nutritional requirements for survival. Usually, normal dietary crude protein in the diet is about 12-14% in the Tibetan sheep production. Recent nutritional guidelines have suggested that supplementation of amino acids in diets were the effective strategies for alleviating low-protein diets induced reduced growth performance in livestock. However, excessive reducing the dietary protein levels make it difficult to meet nutritional requirements for Tibetan sheep, resulting in growth retardation, hypoimmunity, and a higher mortality rate. Therefore, dietary crude protein level is decreased by 2% than the normal dietary crude protein in Tibetan sheep production.
Additionally, detail procedures for analysis chemical composition of diet were added in this manuscript.
L131-139: Are you sure you can detect antioxidant activity in ruminal fluid? Please add the detail steps for how to prepare ruminal fluid to detect these parameters. The same as α -amylase activity.
Response: Thanks for your kind comment. Both antioxidant indicators and digestive enzyme activity of ruminal fluid were detected by enzyme-linked immunosorbent assay. Briefly, the rumen fluids were transferred to polypropylene microtubes and centrifuged at 3000 × g for 20 min at 4 ℃. Then, 10 µL of the supernatants were removed and mixed with 40 µL of diluent. The mixture was vortexed for 10 s and incubated 30 min at 37 ℃. The liquids were removed and then added 50 µL of enzyme-labeled reagent in 96-well plates. The 50 µL of color developing agent color developing agent were vortexed for 10 min at 37 ℃. After termination reaction, the absorbance value were measured under 450 nm wave length by Enzyme labeling analyzer (A51119600C, ThermoFisher, USA).
L147-148: please added the detail information for VFA determine. Such as, GC condition, etc.
Response: Thanks for your kind comment. The detail information of VFA determination were provided.
L180-182: Have all data been tested for normality or not? And how to test?
Response: Thanks for your kind comment. In this study, the KS normality test was performed to estimate data normality.
L195: Some descriptions are incorrect, such as, No significant difference (P<0.05) was observed, please check carefully in the Result section.
Response: Thanks for your kind comment. I am sorry for those mistakes. The descriptions for Tables were rewritten.
L572-583:请在结论中为本手稿添加限制。
回应:感谢您的友好评论。虽然本研究调查了不同Lys/Met比值对低蛋白饲料西藏羊瘤胃菌群丰度和多样性以及重要代谢物合成的影响。然而,需要进一步的研究来准确定义不同CP水平下藏羊日粮中的最佳Lys/Met比率,并更好地量化影响Lys/Met需求的机制。

Round 2
Reviewer 2 Report
Comments and Suggestions for Authors
Thank you for your revisions. I believe you addressed all of my concerns.
Comments on the Quality of English LanguageGenerally good
Reviewer 3 Report
Comments and Suggestions for Authors
The authors have revised according to my suggestion.